# Bacterial diversity on larval and female *Mansonia* spp. from different localities of Porto Velho, Rondonia, Brazil

**Tatiane M. P. Oliveira**[1]*, **Martha V. R. Rojas**[1], **Jandui A. Amorim**[1], **Diego P. Alonso**[1,2], **Dario P. de Carvalho**[3], **Kaio Augusto N. Ribeiro**[3], **Maria Anice Mureb Sallum**[1]

**1** Departamento de Epidemiologia, Faculdade de Saúde Pública, Universidade de São Paulo, São Paulo, SP, Brazil, **2** Instituto de Biotecnologia da UNESP (IBTEC-Campus Botucatu), Botucatu, SP, Brazil, **3** Santo Antônio Energia, Porto Velho, RO, Brazil

\* porangaba@usp.br

## Abstract

Studies based on the bacterial diversity present in *Mansonia* spp. are limited; therefore, the aim of this study was to investigate the bacterial diversity in females and larvae of *Mansonia* spp., describe the differences between the groups identified, and compare the microbiota of larvae from different collection sites. Sequences of the 16S rRNA region from the larvae and females of *Mansonia* spp. were analyzed. Diversity analyzes were performed to verify the possible bacterial differences between the groups and the collection sites. The results showed *Pseudomonas* was the most abundant genus in both females and larvae, followed by *Wolbachia* in females and *Rikenellaceae* and *Desulfovibrio* in larvae. *Desulfovibrio* and *Sulfurospirillum*, sulfate- and sulfur-reducing bacteria, respectively, were abundant on the larvae. *Aminomonas*, an amino acid-degrading bacterium, was found only in larvae, whereas *Rickettsia* was identified in females. Bacterial diversity was observed between females and larvae of *Mansonia* and between larvae from different collection sites. In addition, the results suggest that the environment influenced bacterial diversity.

## Introduction

The genus *Mansonia* Blanchard (1901) is part of the Culicidae family, Mansoniini tribe, which is composed of two subgroups: (1) *Mansonia* with 15 species distributed throughout the New World, and (2) *Mansonioides* with 10 species found in the Old World [1]. Twelve species of *Mansonia* subgenus occur in Brazil [2]. While certain species of *Mansonia* are naturally infected with arboviruses and filariae worms [3], these mosquitoes are not the main carriers of arboviruses in Brazil [4]. The blood-sucking habits of female mosquitoes in the *Mansonia* can lead to discomfort among vertebrate hosts in areas with high mosquito populations. The early life forms have adaptations in the larval siphon and pupal trumpet that enable these mosquitoes to perforate and extract oxygen from the aerenchyma of aquatic macrophyte plants [5, 6].

People have been modifying the natural landscapes for thousands of years, either locally or regionally. Anthropogenic modifications in land cover and land use can cause a notable impact

**Data Availability Statement:** The raw sequences generated and analyzed in this study were

deposited in the Sequence Read Archive (SRA) (PRJNA975891 BioProject).

**Funding:** This work was financially supported by the Research and Development Project from Santo Antônio Energia (ANEEL project CT.PD.124.2018), CNPq grant no. 303382/2022-8, Medical Research Council-São Paulo Research Foundation (FAPESP) CADDE partnership award, MR/S0195/1 and FAPESP 18/14389–0 to MAMS. This study was financed in part by the Coordenação de Aperfeiçoamento de Pessoal de Nível Superior - Brasil (CAPES) - Finance Code 001. The funders had no role in study design, data collection and analysis, decision to publish, or preparation of the manuscript.

**Competing interests:** The authors have declared that no competing interests exist.

on the aquatic ecosystems, and a spatial redistribution of the water resources. Several human changes, including agriculture irrigation, construction of electric power plants, and urbanization, can lead to a reduction in water flow, expand lentic environments, and favor the propagation of aquatic macrophyte plants, proliferation and spread of *Mansonia* spp. [7, 8]. Members of this genus were prevalent in inundated regions along the floodplain of the Solimões-Amazonas Rivers between Tabatinga (Amazonas State) and Ajurixi (Pará state) municipalities, Amazonas state, Brazil, where the macrophyte aquatic plants were ample [9]. Mosquitoes contain a range of symbiotic microorganisms that are crucial to their development [10]. The interactions between mosquitoes and bacteria during larval development can stimulate metabolic changes with functional consequences on adult fitness [11]. In adults, the interactions with bacteria can influence vector competence [12], reproduction, blood digestion, and modulation of pathogenic infections [13, 14].

The bacterial composition in mosquitoes can differ across tissues due to factors such as developmental stage, sex, and environment [15–18]. These microorganisms can be obtained by the larvae from the environment, primarily from water in aquatic habitats, and vertically [19–21]. Bascuñán et al. [22] confirmed that the water samples from breeding sites have more bacterial diversity than the microbiota of larvae, indicating that some bacteria in the water do not have contact with the larvae or cannot survive in their intestine. Much of the bacteria found in the intestine of adult mosquitoes are from the larval environment, and the variation in bacterial diversity between adult mosquitoes and larvae may be because of changes in diet or intestinal restructuring during metamorphosis [22]. The microbiota composition in mosquitoes is highly influenced by their geographic location. This is because each region adds different environmental conditions that can affect the variation of the microbiota, such as the amount of nutrients available in the larval habitat, microclimatic factors, and the availability of blood sources [23–25].

Symbiotic bacteria can be used as a potential tool for both mosquito and pathogen control [26, 27]. As the composition of microbiota in adult mosquitoes is derived from their larval stages and water of the breeding [21], changes in the larval microbiota can impact not only larval development but also adult fitness [10, 11]. Tikhe et al. [27] proved that the use of bacteriophages directed to *Enterobacter* and *Pseudomonas* could reduce the larval development of *Anopheles* and make phage therapy for larval control a viable option. A different approach that uses symbiotic bacteria and is well researched is paratransgenesis. In this process, symbiotic bacteria that have been genetically modified are reintroduced into mosquitoes to produce effector molecules that hinder the development of the pathogen in the host [28]. Despite the importance of symbiotic bacteria in mosquitoes, there are no studies on the bacterial diversity in *Mansonia* spp. Therefore, this study aimed to (1) identify the bacteria present in the females and fourth-instar larvae of *Mansonia* spp. and (2) verify differences in bacterial composition associated with the larvae and females between areas in the vicinities of Santo Antonio Energia hydroelectric power plant (SAE) reservoir in the municipality of Porto Velho, Rondonia state, Brazil.

## Materials and methods

### Sample collection and identification

Larvae and adult mosquitoes were collected in areas between the Jaci-Paraná district and the Porto Velho town in the vicinities of the Santo Antônio Energia Hydroelectric Power Plant (SAE) reservoir in Rondonia State, Brazil (Fig 1). The collection sites are under the influence of the SAE lake in rural and peri-urban communities. Collections were conducted approximately every four months from October 2018 to March 2020 (S2 Table). Adults were collected

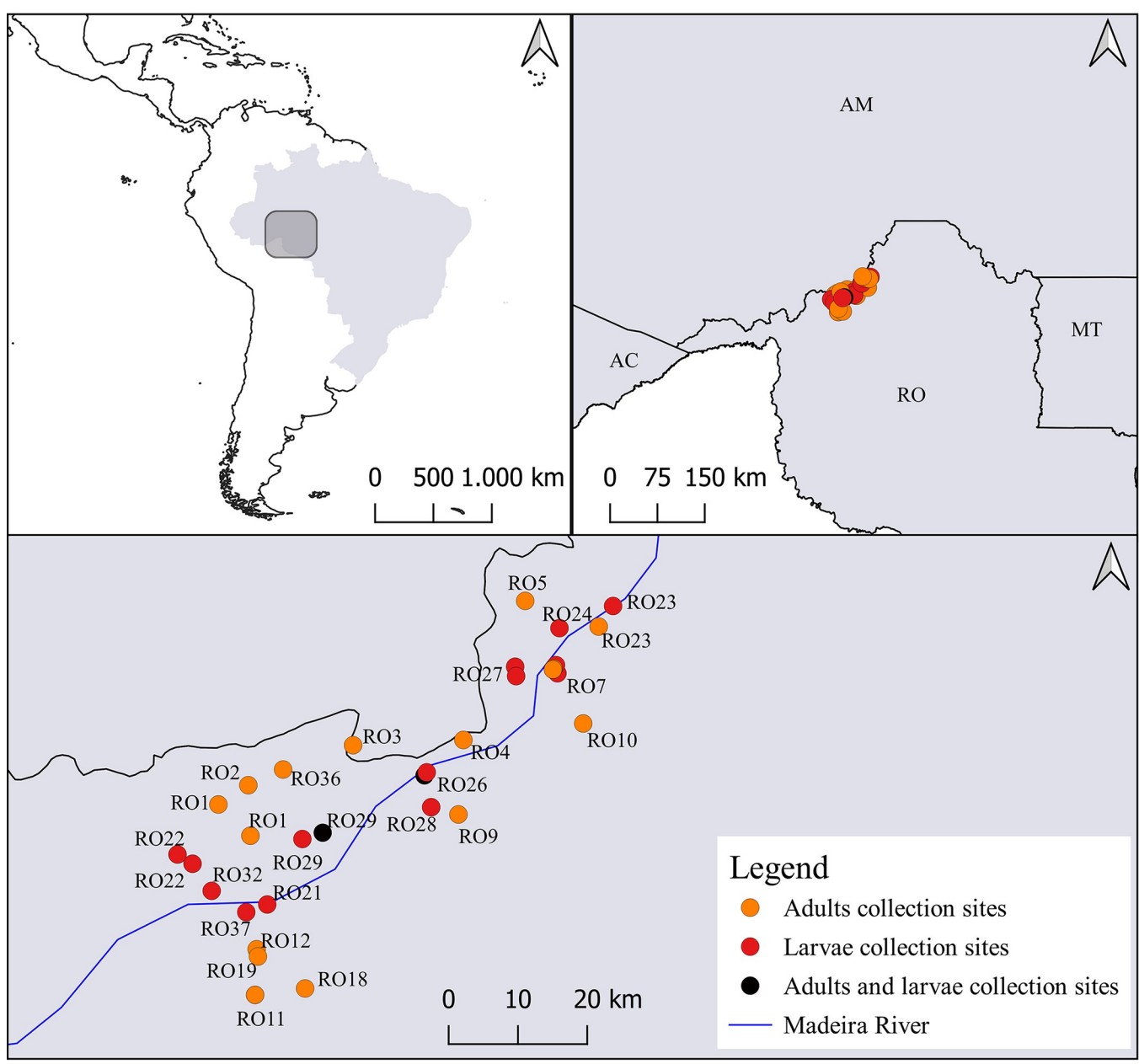

**Fig 1. Sampling locations across the Madeira River region, Porto Velho, Rondonia State, Brazil.** The collection sites in Jaci-Paraná district were RO11, RO18, and RO19 and in Porto Velho town were RO23, RO24, and RO7.

from 18h00 to 06h00 with CDC light traps. Larval collections were carried out in river streams (*Igarapés*) on both edges of the Madeira River (S1 Table). The larvae were collected from the roots of macrophytes aquatic plants at different sites that were chosen randomly among those that were accessible by boat. The roots of *Eichhornia crassipes*, *Limnobium laevigatum*, *Pistia stratiotes*, *Salvinia auriculata*, *Salvinia molesta* and *Salvinia* spp. plants were shaken vigorously in a plastic container filled with clean water get from the collection site to remove the immatures attached. After each collection, the larvae were euthanized in water at 50°C and then were placed in 95% ethanol. The adults were killed with ethyl acetate vapour and kept in small plastic containers with silica gel. Specimens were morphologically identified either to species

or genus levels using Forattini´s identification keys [1]. Females were bisected to separate the abdomen from the head and thorax using the protocol proposed by Nagaki et al. [29]. Only the abdomens of the non-engorged of females and whole fourth-instar larvae were employed in the study. The heads and thoraxes of the females are stored in a -70˚C ultra-freezer for further studies.

## DNA amplification and sequencing

The external surface of each specimen was rinsed with 70% ethanol and ultra-pure water. Genomic DNA was extracted from each specimen using the Quick-DNA Fungal/Bacterial Mini-prep kit (Zymo Research, Irvine, CA, USA) according to the manufacturer's protocol. The DNA was stored at -20˚C until further processing. The V4 fragment of the bacterial 16S rRNA gene was amplified with specific primers [30] linked to Illumina adapter sequences (16S-V4 Forward 5′ `TCGTCGGCAGCGTCAGATGTGTATAAGAGACAGGTGCCAGCMGCCG CGGTAA 3′` and 16S- V4 Reverse 5′ `GTCTCGTGGGCTCGGAGATGTGTATAAGAGACAGG GACTACHVGGGTWTCTAAT 3′`). Each reaction was performed with 1 X GoTaq® Green Master Mix (Promega, Madison, WI, USA), 0.3 μM of each primer, 8 μL of genomic DNA and ultra-pure water to the volume of 20 μL. The amplification of the target region comprised a cycle of 94 ˚C for 3 min followed by 30 cycles of 94 ˚C for 45 s, 55 ˚C for 1 min, 72 ˚C for 1 min, and a final extension of 72 ˚C for 10 min. The PCR products were purified using Agencourt AMPure XP magnetic beads (Beckman Coulter, Brea, CA, USA) and indexed using the Nextera XT Index kit (Illumina, San Diego, CA, USA) according to the manufacturer's instructions. After indexing, the products were purified and quantified by real-time PCR (qPCR) using a KAPA-KK4824 kit (Library Quantification kit, Illumina/Universal), following the manufacturer's recommendations. All samples were normalized to 3 nM and an equimolar pool of DNA was prepared. Sequencing was performed using a MiSeq Reagent Micro v2 kit (300 cycles:2 × 150 base pairs) on a MiSeq sequencer (Illumina).

## Quality control analysis and taxonomy

Illumina paired-end reads were assembled with a minimum overlap of six base pairs using FLASH v. 1.2.11 [31]. Chimeras and low-quality sequences were discarded using Deblur in QIIME2 v.2021-11 software [32]. The steps above were performed with Illumina reads of each run separately, and the sequences and tables were merged using *qiime feature-table merge-seqs* and *qiime feature-table merge* in QIIME2. Sequences of mitochondria, chloroplasts, and archaea were removed using *qiime taxa filter-seqs* in QIIME2. The filtered sequences were used for subsequent analyses. The taxonomy was assigned to QIIME2 using the SILVA 138 database [33, 34].

## Diversity metrics

Diversity metrics (α and β) were generated in QIIME2. The rarefaction curve was visualized with *Ampvis2* package [35] of RStudio v.1.4.1106, which aimed to show the expected number of Amplicon Sequence Variants (ASVs) in each sample for a certain number of sequences. The Mann-Whitney-Wilcoxon Test was performed in RStudio v.1.4.1106 to verify whether the Shannon-Weaver indices were significantly different between larvae and females. Permutal multivariate analysis of variance (PERMANOVA) was performed using QIIME2 with data based on weighted and unweighted UniFrac distances. Values with $p < 0,05$ were considered statistically significant, showing bacterial composition differences between larvae and females and between larvae from different collection sites. The Kruskal-Wallis test, followed by the

Dunn test, was performed to verify whether a given taxonomic group was different between specimens of interest.

## Principal Coordinate Analysis and heatmaps

Principal Coordinate Analysis (PCoA) was performed in RStudio v.1.4.1106 using weighted and unweighted UniFrac phylogenetic distance matrix data generated in QIIME2. The figures generated in the PCoA display the distances observed between the groups in the bacterial communities.

The heatmaps of the bacterial taxa at the genus level display the abundance observed in each taxon per sample/group present in the data from the high-throughput sequencing of the microbiota of females and fourth-instar larvae of *Mansonia* species. Heatmaps were generated for each group using the *qiime2R* package in RStudio, version 1.4.1106.

## Linear Discriminant Analysis

To verify the bacterial diversity in the larvae and females of *Mansonia* species sampled, Linear Discriminant Analysis (LDA) combined with effect size (LEfSe) [36] was employed. This method is widely used to identify biomarkers and was performed with *microbiomeMarker* package [37] in RStudio v.1.4.1106 using the ASV table, SILVA taxonomy, and metadata table. Microbial abundance data were normalized with "CPM" and the score 4 was used as the cutoff value of LDA score. This analysis was also used to find biomarkers between larvae and adults from different collection locations and between different species of *Mansonia*.

## Results

### Species identification

Females assembled for the microbiota analyses were identified as *Mansonia amazonensis*, *Mansonia flaveola*, *Mansonia humeralis*, *Mansonia indubitans*, *Mansonia titillans*, and *Mansonia* spp. (S2 Table). Eighty-nine fourth-instar larvae were collected from *Salvinia auriculata*, *Salvinia* spp. *Eichornia crassipes*, *Pistia stratiotes*, and *Limnobium laevigatum* in 11 collection sites (Table 1). Two hundred and sixty-five females analyzed in this study were gathered from 16 locations. An ID was assigned for each collection site (Fig 1 and Table 1).

### Sequencing data

Three hundred and fifty-four samples were employed to generate bacterial 16S rRNA sequences, amounting to 265 females and 89 fourth-instar larvae. A total of 17,368,746 raw reads (R1 and R2) were produced using four Illumina sequencing runs. The reads of seven samples were excluded during denoising with Deblur in QIIME2, the remaining Illumina 16S reads of 347 samples, amounting to 258 females and 89 larvae (S2 Table and Table 1). After merging the paired end reads and filtering, 5,410,680 sequences were used in the microbiota analyzes (S3 Table).

### Bacterial diversity

In total, 3,535 ASVs corresponding to 667 genera were identified in all samples. *Proteobacteria* was the predominant phylum in both groups (Fig 2). *Pseudomonas* was the most abundant genus in both females and larvae, followed by *Wolbachia* in females and *Rikenellaceae* and *Desulfovibrio* in larvae (Fig 3 and S4 Table). Larvae showed greater bacterial diversity than females (Figs 2 and 3; S1 and S2 Figs).

**Table 1. Number of larvae and females that had the microbiota analyzed by locality and by host plant species.**

| Amount of larvae | ID locality | Host plant species |
|---|---|---|
| 10 (*Mansonia* spp.) | RO21 | *E. crassipes* (6), *Salvinia* spp. (1), and *P. stratiotes* (3) |
| 7 (*Mansonia* spp.) | RO22 | *E. crassipes* (4), *S. auriculata* (2), and *Salvinia* spp. (1) |
| 8 (*Mansonia* spp.) | RO23 | *E. crassipes* (4), *L. laevigatum* (2), *S. auriculata* (1), and *Salvinia* spp. (1) |
| 6 (*Mansonia* spp.) | RO24 | *E. crassipes* (4), *Salvinia* spp. (2) |
| 2 (*Mansonia* spp.) | RO26 | *E. crassipes* (2) |
| 17 (*Mansonia* spp.) | RO27 | *E. crassipes* (17) |
| 2 (*Mansonia* spp.) | RO28 | *E. crassipes* (2) |
| 10 (*Mansonia* spp.) | RO29 | *E. crassipes* (7), *S. auriculata* (3) |
| 10 (*Mansonia* spp.) | RO32 | *E. crassipes* (10) |
| 10 (*Mansonia* spp.) | RO37 | *E. crassipes* (10) |
| 7 (*Mansonia* spp.) | RO7 | *E. crassipes* (6), *S. auriculata* (1) |
| **Amount of females** | **ID locality** | |
| 29 (1: *Ma. flaveola*; 8: *Ma. humeralis*; 8: *Ma. indubitans*; 4: *Ma.* spp.; 8: *Ma. titillans*) | RO1 | |
| 7 (1: *Ma. humeralis*; 1: *Ma. indubitans*; 2: *Ma.* spp.; 3: *Ma. titillans*) | RO10 | |
| 10 (1: *Ma. amazonensis*; 5: *Ma. humeralis*; 1: *Ma. indubitans*; 3: *Ma.* spp.) | RO11 | |
| 26 (4: *Ma. amazonensis*; 16: *Ma. humeralis*; 2: *Ma. indubitans*; 3: *Ma.* spp.; 1: *Ma. titillans*) | RO12 | |
| 8 (2: *Ma. humeralis*; 2: *Ma.* spp.; 4: *Ma. titillans*) | RO18 | |
| 20 (3: *Ma. humeralis*; 8: *Ma. indubitans*; 2: *Ma.* spp.; 7: *Ma. titillans*) | RO19 | |
| 33 (2: *Ma. amazonensis*; 1: *Ma. flaveola*; 3: *Ma. humeralis*; 13: *Ma. indubitans*; 6: *Ma.* spp.; 8: *Ma. titillans*) | RO2 | |
| 14 (8: *Ma. amazonensis*; 6: *Ma. humeralis*) | RO23 | |
| 5 (3: *Ma. amazonensis*; 2: *Ma. humeralis*) | RO26 | |
| 10 (3: *Ma. amazonensis*; 2: *Ma. flaveola*; 4: *Ma. humeralis*; 1: *Ma. indubitans*) | RO29 | |
| 36 (1: *Ma. amazonensis*; 6: *Ma. humeralis*; 2: *Ma. indubitans*; 11: *Ma.* spp.; 16: *Ma. titillans*) | RO3 | |
| 1 (1: *Ma.* spp.) | RO36 | |
| 21 (1: *Ma. amazonensis*; 6: *Ma. humeralis*; 6: *Ma.* spp.; 8: *Ma. titillans*) | RO4 | |
| 1 (1: *Ma.* spp.) | RO5 | |
| 15 (7: *Ma. amazonensis*; 8: *Ma. humeralis*) | RO7 | |
| 22 (3: *Ma. amazonensis*; 1: *Ma. flaveola*; 7: *Ma. humeralis*; 8: *Ma.* spp.; 3: *Ma. titillans*) | RO9 | |

Value inside parentheses in "Host plant species" column shows the number of larvae analyzed from each host plant species in each locality.

## Diversity analysis

The quantity of ASVs in each sample according to the number of sequences is shown in S3 Fig. Diversity metrics were calculated using a normalized dataset. For normalization, a random

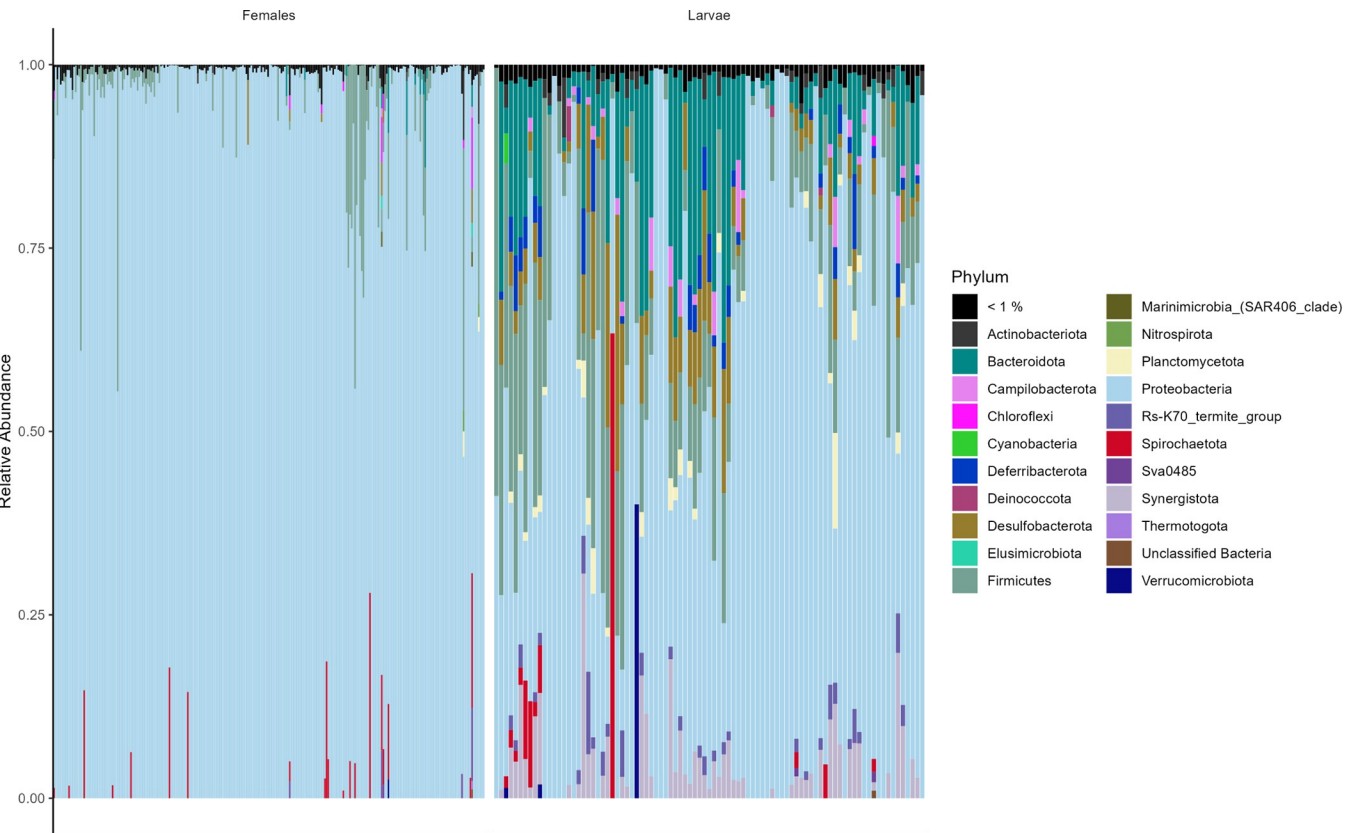

**Fig 2. Bar chart representing the taxonomic composition of phylum-level ASVs in each sample.** The black line represents the phyla that have a relative abundance of less than 1% in each sample.

sub-sampling of 1,491 sequences for each sample was used. This cutoff represented the lowest number of contigs found in the sequenced samples (S3 Table).

The Shannon-Weaver indices ranged from 0.816 to 6.487 between samples (S4 Fig and S5 Table) and did not reveal a normal distribution (Shapiro test; W = 0.84956, p < 0.001). Mann-Whitney-Wilcoxon Test showed the groups differed significantly in α diversity (W = 3815, p < 0.001).

PERMANOVA analyzes (weighted and unweighted UniFrac distances) showed significantly different bacterial compositions between larvae and females (Table 2), and between larvae from different collection sites (S6 Table). Although these differences were statistically significant (p < 0.05), only the difference between the larval and female bacterial compositions were clearly visible in the PCoA (Fig 4 and S5 Fig).

The Kruskal-Wallis test was carried out among samples of larvae from different localities for *Mycobacterium*, *Sulfurospirillum*, *Desulfovibrio*, *Enhydrobacter*, *Aminomonas*, which are associated with the eutrophication process and/or places rich in organic matter. There was a significant difference (p < 0.05) only to *Desulfovibrio*, *Enhydrobacter*, and *Aminomonas*. For these taxa, the Dunn test was performed, and the results are in S7 Table.

## Linear Discriminant Analysis and heatmap

The results of LEfSE showed bacterial differences between larvae and females, with the most abundant bacterial genera in each group having an LDA score of > 4. The genera *Wolbachia*,

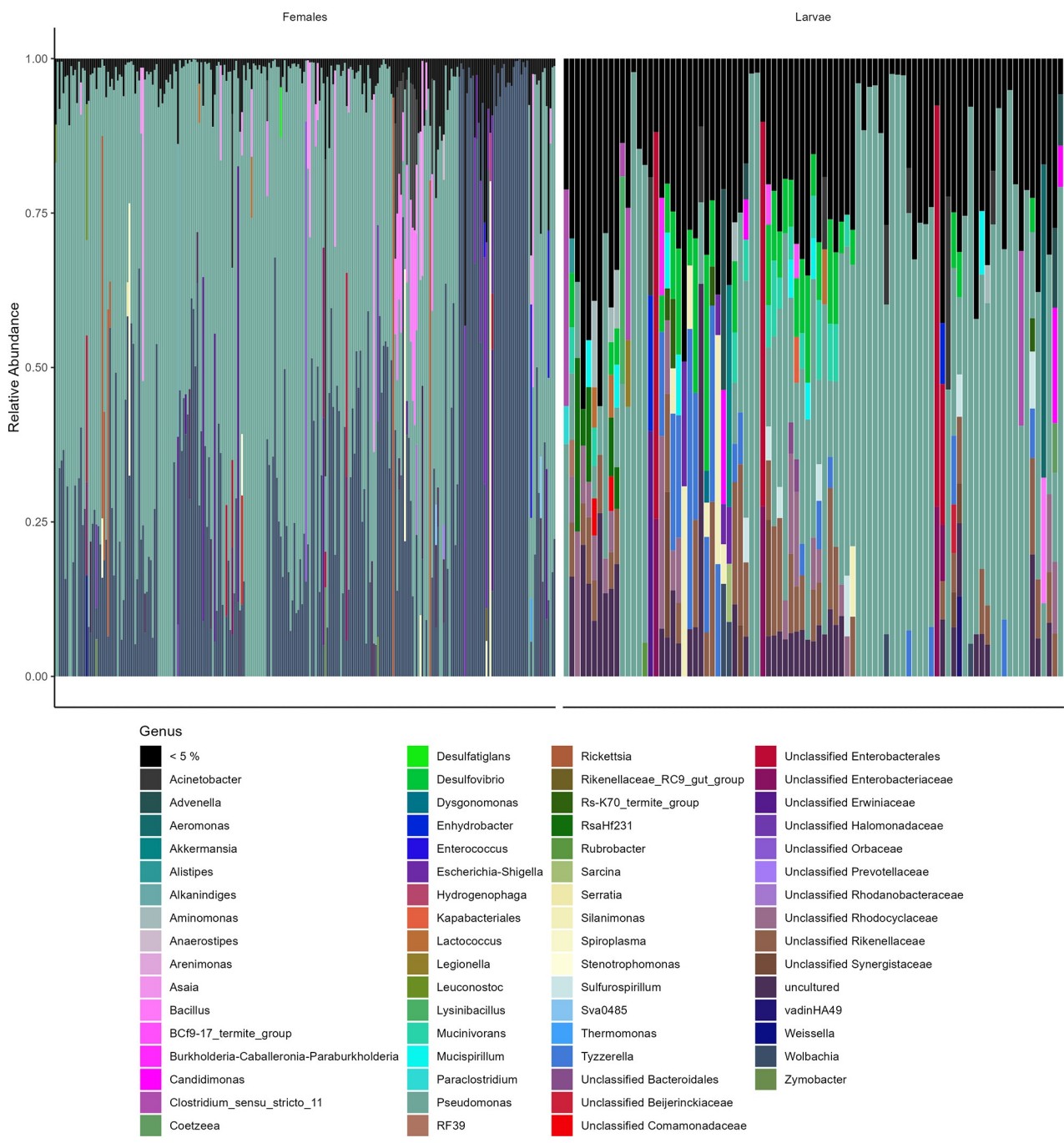

**Fig 3. Bar chart depicting the composition of genus-level taxonomic ASVs in each sample.** The black bar includes the genera that have a relative abundance of less than 5% in each sample.

*Pseudomonas*, *Asaia*, and *Rickettsia* were more abundant in the females than in the larvae (Fig 5). Bacteria of the *Rikenellaceae* and *Rhodocyclaceae* families and *Desulfovibrio* and *Tyzzerella* genera were the most abundant in the larvae. *Rikenellaceae* and *Rhodocyclaceae* families and *Desulfovibrio* and *Tyzzerella* genera showed relative abundances of less than 0,1% in adult mosquitoes, and *Rickettsia* was present only in adult mosquitoes (S4 Table). No taxon was identified as a biomarker between larvae and adults collected from different localities, as well

**Table 2. Result of pairwise permanova of the bacterial composition in larvae and females of *Mansonia* spp.**

| (a) | | Sample size | Permutations | pseudo-F | p-value | q-value |
|---|---|---|---|---|---|---|
| Larvae | Females | 347 | 999 | 109.573 | 0.001 | 0.001 |
| (b) | | Sample size | Permutations | pseudo-F | p-value | q-value |
| Larvae | Females | 347 | 999 | 53.950 | 0.001 | 0.001 |

(a) Weighted UniFrac distance; (b) Unweighted UniFrac distance.

as between different species of *Mansonia*. The heatmap confirmed the previous results for the most abundant genera in each group (S6 Fig).

## Discussion

Comprehending the bacterial communities associated with mosquitoes can assist in creating biological technologies and paratransgenic strategies to control the spread of diseases to humans [38]. Although *Mansonia* mosquitoes are not considered the primary vectors of arboviruses, some were found naturally infected with the Western Equine Encephalitis virus in Argentina [39]. In Brazil, *Mansonia humeralis* was found infected with the Mayaro virus in Rondonia State, Brazil [40], *Mansonia indubitans* and *Mansonia titillans* with Venezuelan Equine Encephalitis virus in Peru [41] and Venezuela [42]. Besides, *Mansonia* (*Mansonioides*) species can be vectors of (1) *Wuchereria bancrofti* in Indonesia [43], Ghana [44], and (2) Rift Valley virus in East Africa [45].

The initial life stages of *Mansonia* species depends on the roots of aquatic macrophytes to get oxygen through the aerenchyma. The bacterial composition can be changed by the environment, especially in the aquatic larval habitat [46], implying that the bacterial communities observed during the larvae stage of *Mansonia* species may vary from those of culicids with distinct ecology. The bacterial genus *Desulfovibrio* contains gram-negative sulfate-reducing bacteria, which are present in organic-rich environments and are linked to water bio-corrosion [47]. This bacterial genus has been identified in the larvae of *Mansonia* spp. studied and other insects [48, 49]. The frequency of *Desulfovibrio* in larvae (3.59%) was higher than in adults (0.005%). Besides *Desulfovibrio*, *Mansonia* spp. larvae hosted a larger relative amount of the sulfur-reducing bacteria of the genus *Sulfurospirillum* [50]. The *Aminomonas* bacteria from

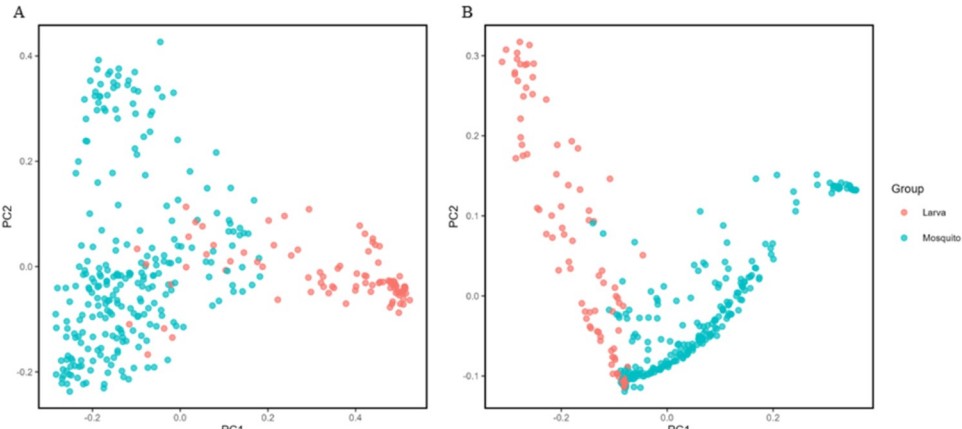

**Fig 4. Principal Coordinate Analysis (PCoA) of the bacterial diversity differences between larvae and females.** (A) PCoA using unweighted UniFrac distance. (B) PCoA using weighted UniFrac distance.

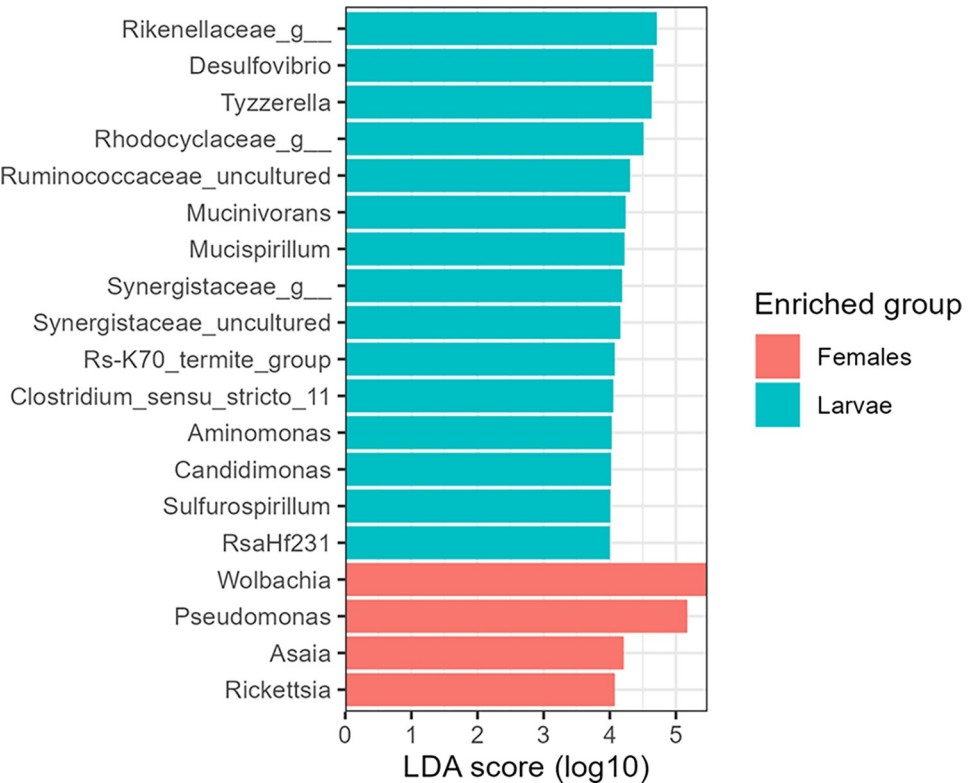

**Fig 5. Result of Linear Discriminant Analysis combined with the effect size (LEfSe) between larvae and females of *Mansonia* spp.**

the *Synergistaceae* family were found in large relative amount in the larvae, however, no contigs were recognized in *Mansonia* females. The *Aminomonas* genus encompasses one identified species, *Aminomonas paucivirans*, which is a microorganism that exclusively degrades amino acids and is strictly anaerobic. It was discovered in an anaerobic lagoon located within a dairy wastewater treatment plant [51]. The vegetation root microhabitat fosters anaerobic bacteria, such as *Aminomonas*.

The deterioration of freshwater ecosystems can be linked to human activities' contamination of nutrients and organic matter [52]. This can lead to a danger to water and food supplies of particular communities and tourism [53]. For this reason, the inspection of dissolved organic matter in water is essential [54]. The eutrophication process is connected to the bacteria of *Enhydrobacter* and *Mycobacterium* genera and *Cyanobacteria* phylum [55–57]. Rojas et al. [58] detected these bacteria in water samples collected across the Madeira River, along the SAE reservoir. The bacterial noted previously was also identified in the larvae samples collected in this study from the same areas examined by Rojas et al. [58]. The findings indicate (1) a correlation between the water habitat and the bacterial makeup of mosquito larvae and (2) the feasible incidence of eutrophication in the areas analyzed.

Like in Rojas et al. [58] variation in bacteria was expected among the larval samples from distinct regions of the same study area. The dissimilarity was demonstrated by the results of PERMANOVA analysis that showed the bacterial diversity in the larvae was considerably different among locations. Other studies exhibited the significant effect of breeding sites on the bacterial communities of mosquitoes during the larval stage [59]. The analysis of statistics revealed variations between larvae from different sites in terms of *Desulfovibrio* and

*Enhydrobacter* taxa (S7 Table). Samples collected from the RO37 region differed from those of RO21, RO23 and RO7 regarding *Desulfovibrio*. Although other locations also have a strong odor because of the decomposition of organic matter, RO37 is close to an extensive field of soybean cultivation. This can lead to an increase in organic matter because of the release of products resulting from this activity or even the dragging of these products by rain, thus explaining the occurrence of this taxon and the difference found. Larvae samples that were collected in the nearest locality (RO7) to a settlement (Vila Teotônio) were the ones that showed a difference with those from the localities RO21, RO27, RO29, RO32 and RO37 in relation to *Enhydrobacter*. The presence of this taxon and the difference found can be explained by the proximity of the collection site to the settlement, which can have led to the increase of organic matter from anthropic activities. Although the larval samples showed bacterial genera associated with eutrophication with significant difference between locations, no bacterial genera that can be a biomarker to monitor organic matter in freshwater was stated in the LEfSe analysis.

The findings of the PERMANOVA analysis revealed that there was a significant statistical difference between the bacterial communities of the studied larvae and females. The detected disparities were analyzed using a PCoA. Also, the larvae possessed greater bacterial diversity than the females did. Variations in bacterial composition between life stages validate the conclusions of other insect studies [60]. Apart from the reduction in bacterial diversity in the pupal stage [19], the microbiota in larvae is anticipated to be more diverse than in adults, possibly because the larval habitat is limited to the roots of floating aquatic macrophytes, which could accumulate organic matter and other residues existing in the aquatic ecosystem. *Eichhornia crassipes*, *Pistia stratiotes*, and *Salvinia molesta* can endure eutrophic aquatic systems that have plenty of sedimentable solids, organic matter, phosphorus, and nitrogen. These plant species can be employed to enhance the limnological parameters of fish farm effluents [61], swine wastewater [62], and to remove impurities from wastewater [63], among other applications. The existence of the aquatic floating plants previously discussed, along with other limnological factors, can enhance the settlement of *Mansonia* species in aquatic ecosystems. Therefore, studies need to be performed to verify if the bacterial composition of *Mansonia* larvae can serve as an alternative indicator of water quality when floating macrophytes are abundant.

It is largely known that geographic location, aquatic ecosystem biotic and abiotic factors, and mosquito species can influence on the composition of the microbiota [23, 25, 59]. Thus, this study faces a limitation in understanding the bacterial difference found between the adults and larvae because the later life stages were identified to the genus level. Likely, geographic location and the mosquito species may have contributed to the differences observed in the bacterial composition of the larvae and females.

It is worth noting the *Rickettsia* in females, a microorganism group that includes species linked to spotted fever. *Rickettsia rickettsia* and *Rickettsia parkeri* [64] can cause rickettsiosis in Brazil. The examination of the sequence revealed that six females had many contigs (> 1000) of this bacterium (S4 Table). This result indicates that mosquitoes might have blood-fed on a *Rickettsia* reservoir (e.g., capybara). Despite the females analyzed were not visibly swollen, traces of blood that remained in the abdomen were high-throughput sequenced, allowing the identification of a microorganism that is not commonly observed in mosquitoes. Further studies will be required to confirm *Rickettsia* in *Mansonia*.

Estimated that approximately 65% of the insect species are infected with *Wolbachia* [65]. This endosymbiotic bacterium has been extensively studied to be employed in biocontrol programs [66]. It influences pathogen transmission, blood feeding and host mosquito reproduction, being able to induce parthenogenesis, male killing, feminization, and cytoplasmic incompatibility (CI) [67–70]. The cross sterility generated by CI results in a reduction in the mosquito population and a decrease in the number of vector-borne disease cases [69, 71]. The

excellent results of the *Wolbachia* method for dengue control in different regions of the world [72] points the potential of the use of this bacterium to control vector-borne diseases. This bacterium was present in the larvae and females analyzed and was more abundant in the latter stage. *Wolbachia* was previously identified in *Mansonia uniformis* and *Mansonia annulifera* females collected from the Alappuzha District, Kerala State, India [73].

*Asaia* is abundant in the salivary glands, reproductive organs, and intestines of different mosquito species [74–76] and can colonize other insects [77]. Because of its characteristics, such as stable association, the ability to colonize different tissues, vertical transmission, cultivation, and genetic manipulation, this bacterium is a candidate for paratransgenic malaria control [78]. In the present study, *Asaia* was abundant in the field collected *Mansonia* females. Further studies are needed to verify the *Mansonia* tissues that this bacterium colonizes.

In conclusion, differences in bacterial composition were observed between the larvae and females of *Mansonia* mosquitoes and between different collection sites, showing that the environment influences this diversity. *Wolbachia* was found in both life stages of *Mansonia*. No taxon present in larval samples indicated a biomarker for monitoring the level of pollution of the larval habitats.

## Supporting information

**S1 Fig. Bar graph of phylum-level taxonomic amplicon sequence variants (ASV) composition in each sample.** The black bar comprises the phyla that have a relative abundance of less than 1% in each sample. The sample corresponding to each bar is indicated.
(PNG)

**S2 Fig. Bar graph of genus-level taxonomic amplicon sequence variants (ASV) composition in each sample.** The black bar comprises the genera that have a relative abundance of less than 5% in each sample. The sample corresponding to each bar is indicated.
(PNG)

**S3 Fig. Rarefaction curve.** ASVs count per given sequencing depth in each sample.
(PDF)

**S4 Fig. Box plot of Shannon indexes of each group (larvae and females).**
(PDF)

**S5 Fig. Principal Coordinate Analysis (PCoA) of the bacterial diversity differences between different larvae collection sites.** (A) PCoA using unweighted Unifrac distance. (B) PCoA using weighted Unifrac distance.
(PDF)

**S6 Fig. Heatmap of sequences with taxonomic assignment to genus level.** The color gradient (yellow to purple) represents relative abundance. Yellow: higher bacterial abundance. Purple: lowest bacterial abundance.
(PDF)

**S1 Table. Description of larval collection sites.**
(XLSX)

**S2 Table. Specimens used in the metabarcoding analyses and respective information about collection date, species, and geographic coordinates of the collection.**
(XLS)

**S3 Table. Count 16S rRNA contigs in each sample after filtering steps.**
(XLSX)

**S4 Table. Count of 16S rRNA contigs of each bacterial genus in each sample.**
(XLSX)

**S5 Table. Shannon index of each sample.**
(DOCX)

**S6 Table. Result of pairwise permanova of the bacterial composition in larvae in relation to collection site.**
(DOCX)

**S7 Table. Result of the Dunn test for *Desulfovibrio*, *Enhydrobacter* and *Aminomas* in larvae samples collected at different localities.**
(DOCX)

## Author Contributions

**Conceptualization:** Martha V. R. Rojas, Maria Anice Mureb Sallum.

**Data curation:** Tatiane M. P. Oliveira.

**Formal analysis:** Tatiane M. P. Oliveira.

**Funding acquisition:** Maria Anice Mureb Sallum.

**Investigation:** Martha V. R. Rojas, Jandui A. Amorim, Diego P. Alonso, Dario P. de Carvalho, Kaio Augusto N. Ribeiro, Maria Anice Mureb Sallum.

**Methodology:** Martha V. R. Rojas, Maria Anice Mureb Sallum.

**Project administration:** Maria Anice Mureb Sallum.

**Supervision:** Maria Anice Mureb Sallum.

**Writing – original draft:** Tatiane M. P. Oliveira, Maria Anice Mureb Sallum.

**Writing – review & editing:** Tatiane M. P. Oliveira, Maria Anice Mureb Sallum.

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
