## [Decision Letter · Decision Letter 0]

14 Jul 2023

PONE-D-23-18212Bacterial diversity on larval and adult female Mansonia spp. from different localities of Porto Velho, Rondonia, Brazil.PLOS ONE

Dear Dr. Oliveira,

Thank you for submitting your manuscript to PLOS ONE. After careful consideration, we feel that it has merit but does not fully meet PLOS ONE’s publication criteria as it currently stands. Therefore, we invite you to submit a revised version of the manuscript that addresses the points raised during the review process.

We look forward to receiving your revised manuscript.

Kind regards,

Ahmed Ibrahim Hasaballah

Academic Editor

PLOS ONE

Journal Requirements:

   "This work was supported by the Research and Development Project from Santo Antônio Energia (ANEEL project CT.PD.124.2018), and CNPq grant no. 303382/2022-8 to MAMS."

6. We note that Figure 1 in your submission contain map/satellite images which may be copyrighted. All PLOS content is published under the Creative Commons Attribution License (CC BY 4.0), which means that the manuscript, images, and Supporting Information files will be freely available online, and any third party is permitted to access, download, copy, distribute, and use these materials in any way, even commercially, with proper attribution. For these reasons, we cannot publish previously copyrighted maps or satellite images created using proprietary data, such as Google software (Google Maps, Street View, and Earth). For more information, see our copyright guidelines: http://journals.plos.org/plosone/s/licenses-and-copyright.

Reviewers' comments:

Reviewer's Responses to Questions

**Comments to the Author**

1. Is the manuscript technically sound, and do the data support the conclusions?

Reviewer #1: Partly

Reviewer #2: Partly

2. Has the statistical analysis been performed appropriately and rigorously? 

Reviewer #1: I Don't Know

Reviewer #2: Yes

3. Have the authors made all data underlying the findings in their manuscript fully available?

Reviewer #1: No

Reviewer #2: Yes

4. Is the manuscript presented in an intelligible fashion and written in standard English?

Reviewer #1: Yes

Reviewer #2: Yes

5. Review Comments to the Author

Reviewer #1: The manuscript titled “ Bacterial diversity on larval and adult female Mansonia spp. from different localities of Porto Velho, Rondonia, Brazil” describes bacterial diversity in field collected Mansonia spp. from Brazil.

My comments are as follows.

1. Overall, the manuscript needs some language editing. Although it’s not a major issue. I highly encourage the authors to have it checked by a language editor.

2. The following sentence needs to be rephrased “Species of this genus can be abundant in environments that are affected by changes in land use.”

3. It is unclear what the authors are implying here “Mansonia spp. are mosquito communities that are abundant along the Amazon Solimões River basin.” Please clarify this sentence.

4. The rationale behind studying the difference between life stages and locations is not clearly stated in the introduction.

5. Line 70- “Morphological identification of the specimens at the species level was performed using Forattini’s identification key” is this for adults or larvae ?

6. The larvae were not identified beyond genus level. However, the adult females were. The authors need to use the extracted DNA to sequence and identify the larvae at species level. Mosquito larvae of different species can have significant differences in their microbiota even if they are reared in the same breeding site. Thus, treating all the larvae as Mansonia spp. and comparing them to adult females of different Mansonia species is not a good variable for the microbiota analyses.

7. What does the figure legend <0.1% mean ? The resolution and color scheme of the figures needs a lot of improvement.

8. Authors need to mention that mosquito microbiota can serve as a potential tool for mosquito control. For example, paratransgenesis for adult mosquitoes and phage therapy for larval mosquitoes.

9. There is no discussion about Wolbachia and its use for mosquito control and blocking arbovirus transmission.

Reviewer #2: The present manuscript, "Bacterial diversity on larval and adult female Mansonia spp. from different localities of Porto Velho, Rondonia, Brazil" is a research article that investigated the bacterial diversity in the microbiota of mosquitoes collected around Santo Antonio Energia Hydroelectric Power Plant (SAE) reservoir. The results indicated significant differences between larvae and adult insects. Although the information provided here is relevant and must be disclosed, some points of the methods need to be clarified (detailed bellow) and data could be deeper explored. Moreover, the motivation of the study beyond the "importance of the microbiota” is not clear. In the following text I present some comments, questions and suggestions about the present manuscript.

Introduction:

In “Species of this genus can be abundant in environments that are affected by changes in land use” (lines 38-39): please specify how land use can impact the ecology of Mansonia species.

The term “Mansonia communities” is unusual. Since in ecology the word “community” describe a group of interacting species living in the same location, one could imagine a “community” formed only by species of this genus (what would be unreal in the natural environment). I suggest Mansonia aggregates or colonies.

Line 54, In “Despite its importance”: Do the authors refer to the importance of the microbiota or of Mansonia? Please clarify.

Why did the authors choose areas close to the hydroelectric plant to sample mosquitoes? How has this environment been impacted by human action? What types of breeding sites were sampled? How were they impacted by the installation of the hydroelectric plant? The motivation of the study beyond the "importance" of the microbiota is not clear.

Methods:

How were larvae and adults collected? Please clarify.

Please indicate locality IDs in the map. It is also necessary to indicate which collection sites were impacted by human activity, by the installation of the hydroelectric plant or have other characteristics.

Why larvae and adults were not collected in the same places? Collection in different sites could contribute to differences seen in the microbiota of different life stages.

Results:

Please indicate in the manuscript how many larvae per collection site/plant host species had the microbiota analyzed. In the same way, inform how many female mosquitoes were processed per species/collection site. This information could be presented in a table.

The meaning of “The number attributed to the points reflects all the locations sampled in the region. Although the numbers were not sequential, larvae collection was carried at 11 points only” (lines 144-146) is not clear.

It is well known in the literature that immature and adult mosquitoes have different microbiotas (as said in line 247). However, the sequencing data generated here can be better explored. For example, there were any differences in bacterial diversity or composition between plant species, mosquito species, collection site or any other characteristic of land use? It must be also considered in ecological analysis.

Figure 2 and 3: please indicate in the x-axis mosquito species (if known) and collection site).

Discussion:

I am not sure if the affirmation “Its [Desulfovibrio] presence in the bacterial composition of larvae collected at different sites confirms that the environment contributes to the larval bacterial composition” (lines 218-220) is supported by the study results. Were all collection sites similar to each other and rich in organic matter?

Lines 234-242: the authors subjectively suggest that some bacteria genera (Desulfovibrio, Sulfurospirillum and Aminomonas) would be characteristic of mosquito larvae from places rich in organic matter. What are the differences between RO32 collection site and other regions, such as RO37, RO29, RO27, and RO24? What is the meaning of “smaller number of contigs” (line 237)? Is it a significant difference? The same analyzes applied to show differences between larvae and females could be used to detect differences in the microbiota of mosquitoes from distinct environments.

In “Further studies are required to verify whether these genera can be used as biomarkers to monitor organic matter in freshwater” (lines 241-242) - perhaps the present study could apply analyzes (such as LDA and LEfSe) to detect differentially abundant bacteria (biomarkers) between areas with different classes of eutrophication.

In the same way, the affirmation “However, the bacterial composition Mansonia larvae can serve as an alternative indicator of water quality when floating macrophytes are abundant” (lines 259-261) needs a better support from the study data - how did larvae microbiota vary according to water quality?

In “Despite the lack of differences between the bacterial compositions of larvae from different breeding sites in the PCoA” (lines 262-263) - a better description of collection sites should be presented, describing similarities between breeding sites. This information should be considered in statistical analysis. It may help to understand if microbiota diversity and composition vary according to some environmental characteristic, such as degrees of human impact or presence of areas flooded by the hydroelectric plant, etc.

Collection in different sites could contribute to differences seen in the microbiota of different life stages and must be considered in the discussion.

6. PLOS authors have the option to publish the peer review history of their article (what does this mean?). If published, this will include your full peer review and any attached files.

Reviewer #1: No

Reviewer #2: No

---

## [Author Response · Author response to Decision Letter 0]

28 Aug 2023

Reviewer #1

1. Overall, the manuscript needs some language editing. Although it’s not a major issue. I highly encourage the authors to have it checked by a language editor.

Thank you for your comment. The language has been checked and corrections have been performed to the text.

2. The following sentence needs to be rephrased “Species of this genus can be abundant in environments that are affected by changes in land use.”

Thank you for this observation. The text has been changed to: “People have been modifying the natural landscapes for thousands of years, either locally or regionally. Anthropogenic modifications in land cover and land use can cause a notable impact on the aquatic ecosystems, and a spatial redistribution of the water resources. Several human changes, including agriculture irrigation, construction of electric power plants, and urbanization, can lead to a reduction in water flow, expand lentic environments, and favor the propagation of aquatic macrophyte plants, proliferation and spread of Mansonia spp. [7,8].” (Lines 40-46).

3. It is unclear what the authors are implying here “Mansonia spp. are mosquito communities that are abundant along the Amazon Solimões River basin.” Please clarify this sentence.

Thank you for this comment. We clarified this sentence and now reads are: “Members of this genus were prevalent in inundated regions along the floodplain of the Solimões-Amazonas Rivers between Tabatinga (Amazonas State) and Ajurixi (Pará state) municipalities, Amazonas state, Brazil, where the macrophyte aquatic plants were ample [9].” (Lines 46-49).

4. The rationale behind studying the difference between life stages and locations is not clearly stated in the introduction.

Thank you for this comment. We added the following text to clarify possible differences in the composition of the microbiota between samples from different locations and different stages of development: “Bascuñán et al. [22] confirmed that the water samples from breeding sites have more bacterial diversity than the microbiota of larvae, indicating that some bacteria in the water do not have contact with the larvae or cannot survive in their intestine. Much of the bacteria found in the intestine of adult mosquitoes are from the larval environment, and the variation in bacterial diversity between adult mosquitoes and larvae may be because of changes in diet or intestinal restructuring during metamorphosis [22]. The microbiota composition in mosquitoes is highly influenced by their geographic location. This is because each region adds different environmental conditions that can affect the variation of the microbiota, such as the amount of nutrients available in the larval habitat, microclimatic factors, and the availability of blood sources [23-25].” (Lines 57-66).

5. Line 70- “Morphological identification of the specimens at the species level was performed using Forattini’s identification key” is this for adults or larvae ?

Thank you for this observation. The Forattini’s identification key was used for all specimens. The text was changed to: “Specimens were morphologically identified either to species or genus levels using Forattini´s identification keys [1].” (Lines 98-99).

6. The larvae were not identified beyond genus level. However, the adult females were. The authors need to use the extracted DNA to sequence and identify the larvae at species level. Mosquito larvae of different species can have significant differences in their microbiota even if they are reared in the same breeding site. Thus, treating all the larvae as Mansonia spp. and comparing them to adult females of different Mansonia species is not a good variable for the microbiota analyses.

Thank you for this comment. We agreed on the importance of identifying the larvae at the species level, however, unfortunately we were unable to carry out this step, as there is no more genetic material from the larval specimens. Analyzes of the bacterial diversity between larvae and adults were carried out considering only the genus, without considering the species of the adults.

7. What does the figure legend <0.1% mean ? The resolution and color scheme of the figures needs a lot of improvement.

Thank you for this observation. The values 1% and 0.1% shown in figures 2 and 3 indicate taxa with median relative abundance of less than 1% and 0.1%, respectively. The figures 2 and 3 were redone and have better resolution. The values 1% and 5% present in the new figures represent phyla that have a relative abundance of less than 1% in each sample (Figure 2) and genera that have a relative abundance of less than 5% in each sample (Figure 3), respectively. We added the following information in legend of the figure 2: “Bar chart representing the taxonomic composition of phylum-level ASVs in each sample. The black line represents the phyla that have a relative abundance of less than 1% in each sample.” (Lines 203-205) and in the legend of the figure 3: “Bar chart depicting the composition of genus-level taxonomic ASVs in each sample. The black bar includes the genera that have a relative abundance of less than 5% in each sample.” (Lines 206-208).

8. Authors need to mention that mosquito microbiota can serve as a potential tool for mosquito control. For example, paratransgenesis for adult mosquitoes and phage therapy for larval mosquitoes.

Thank you for this comment. We add the following information: “Symbiotic bacteria can be used as a potential tool for control of both mosquito and of pathogen transmitted by mosquito vector [26,27]. As the composition of microbiota in adult mosquitoes is derived from their larval stages and water of the breeding [21], changes in the larval microbiota impact not only larval but also adult development. Tikhe et al. [27] proved that the use of bacteriophages directed to Enterobacter and Pseudomonas could reduce the larval development of Anopheles and make phage therapy for larval control a viable option. A different approach that uses symbiotic bacteria and is well researched is paratransgenesis. In this process, symbiotic bacteria that have been genetically modified are reintroduced into mosquitoes to produce effector molecules that hinder the development of the pathogen in the host [28].” (Lines 67-76).

9. There is no discussion about Wolbachia and its use for mosquito control and blocking arbovirus transmission.

Thank you for this observation. We add the following information in the text: “It influences pathogen transmission, blood feeding and host mosquito reproduction, being able to induce parthenogenesis, male killing, feminization, and cytoplasmic incompatibility (CI) [67-70]. The cross sterility generated by CI results in a reduction in the mosquito population and a decrease in the number of vector-borne disease cases [69,71]. The excellent results of the Wolbachia method for dengue control in different regions of the world [72] points the potential of the use of this bacterium to control vector-borne diseases.” (Lines 336-342).

Reviewer #2

In “Species of this genus can be abundant in environments that are affected by changes in land use” (lines 38-39): please specify how land use can impact the ecology of Mansonia species.

Thank you for this observation. The text has been changed to: “People have been modifying the natural landscapes for thousands of years, either locally or regionally. Anthropogenic modifications in land cover and land use can cause a notable impact on the aquatic ecosystems, and a spatial redistribution of the water resources. Several human changes, including agriculture irrigation, construction of electric power plants, and urbanization, can lead to a reduction in water flow, expand lentic environments, and favor the propagation of aquatic macrophyte plants, proliferation and spread of Mansonia spp. [7,8].” (Lines 40-46).

The term “Mansonia communities” is unusual. Since in ecology the word “community” describe a group of interacting species living in the same location, one could imagine a “community” formed only by species of this genus (what would be unreal in the natural environment). I suggest Mansonia aggregates or colonies.

Thank you for this suggestion. The term “Mansonia communities” has been removed from the text.

Line 54, In “Despite its importance”: Do the authors refer to the importance of the microbiota or of Mansonia? Please clarify.

Thank you for this comment. In “Despite its importance” we were referring to the importance of the microbiota. To make it clear to the reader, the sentence above was changed to “Despite the importance of symbiotic bacteria in mosquitoes, there are no studies on the bacterial diversity in Mansonia spp.” (Lines 76-78).

Why did the authors choose areas close to the hydroelectric plant to sample mosquitoes? How has this environment been impacted by human action? What types of breeding sites were sampled? How were they impacted by the installation of the hydroelectric plant? 

Thank you for this comment. To clarify the above issues, we added a table supplementary (S1 Table) with information about each larval collection site and the following text has been added: “The collection sites are under the influence of the SAE lake in rural and peri-urban communities. Adults were collected from 18h00 to 06h00 with CDC light traps. Larval collections were carried out in river streams (Igarapés) on both edges of the Madeira River (S1 Table), from October 2018 to March 2020 (S2 Table).” (Lines 87-91).

The motivation of the study beyond the "importance" of the microbiota is not clear.

Thank you for this observation. The text was changed to clear the importance of the microbiota in this study and now it reads: “Symbiotic bacteria can be used as a potential tool for control of both mosquito and of pathogen transmitted by mosquito vector [26,27]. As the composition of microbiota in adult mosquitoes is derived from their larval stages and water of the breeding [21], changes in the larval microbiota impact not only larval but also adult development. Tikhe et al. [27] proved that the use of bacteriophages directed to Enterobacter and Pseudomonas could reduce the larval development of Anopheles and make phage therapy for larval control a viable option. A different approach that uses symbiotic bacteria and is well researched is paratransgenesis. In this process, symbiotic bacteria that have been genetically modified are reintroduced into mosquitoes to produce effector molecules that hinder the development of the pathogen in the host [28].” (Lines 67-76).

How were larvae and adults collected? Please clarify.

Thank you for this comment. The following text provides further details on collecting larvae and adults: “Larvae and adult mosquitoes were collected in areas between the Jaci-Paraná district and the Porto Velho town in the vicinities of the Santo Antônio Energia Hydroelectric Power Plant (SAE) reservoir in Rondonia State, Brazil (Fig 1). The collection sites are under the influence of the SAE lake in rural and peri-urban communities. Adults were collected from 18h00 to 06h00 with CDC light traps. Larval collections were carried out in river streams (Igarapés) on both edges of the Madeira River (S1 Table), from October 2018 to March 2020 (S2 Table). The larvae were collected from the roots of macrophytes aquatic plants at different sites that were chosen randomly among those that were accessible by boat. The roots of Eichhornia crassipes, Limnobium laevigatum, Pistia stratiotes, Salvinia auriculata, Salvinia molesta and Salvinia spp. plants were shaken vigorously in a plastic container filled with clean water get from the collection site to remove the immatures attached. After each collection, the larvae were euthanized in water at 50 ºC and then were placed in 95% ethanol. The adults were killed with ethyl acetate vapour and kept in small plastic containers with silica gel.” (Lines 85-98).

Please indicate locality IDs in the map. It is also necessary to indicate which collection sites were impacted by human activity, by the installation of the hydroelectric plant or have other characteristics.

Thank you for this suggestion. The figure 1 has been replaced by another one that shows the ID localities. To clear the question above, we also added information about larval collection sites in S1 Table. 

The figura 1 legend was changed to: “Sampling locations across the Madeira River region, Porto Velho, Rondonia State, Brazil. The collection sites in Jaci-Paraná district were RO11, RO18, and RO19 and in Porto Velho town were RO23, RO24, and RO7.” (Lines 104-106).

Why larvae and adults were not collected in the same places? Collection in different sites could contribute to differences seen in the microbiota of different life stages.

Thank you for this observation. All collections were carried out between the Jaci-Paraná district and the Porto Velho town, but not all larvae and adults collections were carried out at the same point due to the inaccessibility of the boat in some areas.

Please indicate in the manuscript how many larvae per collection site/plant host species had the microbiota analyzed. In the same way, inform how many female mosquitoes were processed per species/collection site. This information could be presented in a table.

Thank you for this suggestion. We added this information in the Table 1 (lines 184-187).

The meaning of “The number attributed to the points reflects all the locations sampled in the region. Although the numbers were not sequential, larvae collection was carried at 11 points only” (lines 144-146) is not clear.

Thank you for this observation. To make it clearer to the reader, we changed the sentence to: “Two hundred and sixty-five females analyzed in this study were gathered from 16 locations. An ID was assigned for each collection site (Fig 1 and Table 1).” (Lines 181-183).

It is well known in the literature that immature and adult mosquitoes have different microbiotas (as said in line 247). However, the sequencing data generated here can be better explored. For example, there were any differences in bacterial diversity or composition between plant species, mosquito species, collection site or any other characteristic of land use? It must be also considered in ecological analysis.

Thank you for this comment. We added to the study a table with the description of each larvae collection site (S1 Table), as well as added LEfSE analysis between larvae from different localities and between different species of Mansonia adults. It was not possible to carry out LEfSE between larvae collected from different species of macrophytes due to error in analysis because of the small number of samples or the small variation of feture abundances within a given group. The Kruskal-Wallis test was also added to verify if larvae from different localities showed differences in relation to some taxa associated to eutrophication process. 

Thus, we added to the study the LEfSE analysis between females and larvae from different localities and between different species of Mansonia adults: “This analysis was also used to find biomarkers between larvae and adults from different collection locations and between different species of Mansonia.” (Lines 166-168) and as a result we add: “No taxon was identified as a biomarker between larvae and adults collected from different localities, as well as between different species of Mansonia.” (Lines 244-246).

We also added information about the Kruskal-Wallis test: “The Kruskal-Wallis test, followed by the Dunn test, was performed to verify whether a given taxonomic group was different between specimens of interest.” (Lines 148-150) and as a result we add: “The Kruskal-Wallis test was carried out among samples of larvae from different localities for Mycobacterium, Sulfurospirillum, Desulfovibrio, Enhydrobacter, Aminomonas, which are associated with the eutrophication process and/or places rich in organic matter. There was a significant difference (p < 0.05) only to Desulfovibrio, Enhydrobacter, and Aminomonas. For these taxa, the Dunn test was performed, and the results are in S7 Table.” (Lines 223-227).

We added the following information in the discussion: “Like in Rojas et al. [58] variation in bacteria was expected among the larval samples from distinct regions of the same study area. The dissimilarity was demonstrated by the results of PERMANOVA analysis that showed the bacterial diversity in the larvae was considerably different among locations. Other studies exhibited the significant effect of breeding sites on the bacterial communities of mosquitoes during the larval stage [59]. The analysis of statistics revealed variations between larvae from different sites in terms of Desulfovibrio and Enhydrobacter taxa (S7 Table). Samples collected from the RO37 region differed from those of RO21, RO23 and RO7 regarding Desulfovibrio. Although other locations also have a strong odor because of the decomposition of organic matter, RO37 is close to an extensive field of soybean cultivation. This can lead to an increase in organic matter because of the release of products resulting from this activity or even the dragging of these products by rain, thus explaining the occurrence of this taxon and the difference found. Larvae samples that were collected in the nearest locality (RO7) to a settlement (Vila Teotônio) were the ones that showed a difference with those from the localities RO21, RO27, RO29, RO32 and RO37 in relation to Enhydrobacter. The presence of this taxon and the difference found can be explained by the proximity of the collection site to the settlement, which can have led to the increase of organic matter from anthropic activities. Although the larval samples showed bacterial genera associated with eutrophication with significant difference between locations, no bacterial genera that can be a biomarker to monitor organic matter in freshwater was stated in the LEfSe analysis.” (Lines 288-307).

Figure 2 and 3: please indicate in the x-axis mosquito species (if known) and collection site).

 Thank you for this suggestion. We insert two figures (S5 Fig and S6 Fig) of relative abundance of genera and phyla and that indicate the samples in the x-axis. Thus, with the sample ID, the reader will not only have information on the locality ID and species of the mosquito, but also on all the other data show in S2 Table.

I am not sure if the affirmation “Its [Desulfovibrio] presence in the bacterial composition of larvae collected at different sites confirms that the environment contributes to the larval bacterial composition” (lines 218-220) is supported by the study results. Were all collection sites similar to each other and rich in organic matter?

Thank you for this observation. We agree that our results do not support this statement, because the fact that this genus can be found in environments rich in organic matter, we do not have information on its presence in the larval habitat of the studied locations. So, this sentence was removed from the text. Information about each larval collection site has been added in S1 Table.

About the environment contributing to the bacterial composition of the larvae, the following text was added: “The eutrophication process is connected to the bacteria of Enhydrobacter and Mycobacterium genera and Cyanobacteria phylum [55-57]. Rojas et al. [58] detected these bacteria in water samples collected across the Madeira River, along the SAE reservoir. The bacterial noted previously was also identified in the larvae samples collected in this study from the same areas examined by Rojas et al. [58]. The findings indicate (1) a correlation between the water habitat and the bacterial makeup of mosquito larvae and (2) the feasible incidence of eutrophication in the areas analyzed.” (Lines 280-287).

Lines 234-242: the authors subjectively suggest that some bacteria genera (Desulfovibrio, Sulfurospirillum and Aminomonas) would be characteristic of mosquito larvae from places rich in organic matter. What are the differences between RO32 collection site and other regions, such as RO37, RO29, RO27, and RO24? What is the meaning of “smaller number of contigs” (line 237)? Is it a significant difference? The same analyzes applied to show differences between larvae and females could be used to detect differences in the microbiota of mosquitoes from distinct environments.

Thank you for your observation. The collections of immatures were carried out in localities of influence of the SAE and information about each one of these localities was added in S1 Table. We agree that it is really not ideal to consider the differences between these taxa only by the number of contigs. Despite of the PERMANOVA analysis showed that there is difference in bacterial diversity between larvae from different collection sites, we had not carried out a DA analysis or another analysis to find out if larva sampled from one locality could be different in relation to some of these taxa.

Thus, we added to the study the LEfSE analysis between females and larvae from different localities and between different species of Mansonia adults: “This analysis was also used to find biomarkers between larvae and adults from different collection locations and between different species of Mansonia.” (Lines 166-168) and as a result we add: “No taxon was identified as a biomarker between larvae and adults collected from different localities, as well as between different species of Mansonia.” (Lines 244-246).

We also added information about the Kruskal-Wallis test: “The Kruskal-Wallis test, followed by the Dunn test, was performed to verify whether a given taxonomic group was different between specimens of interest.” (Lines 148-150) and as a result we add: “The Kruskal-Wallis test was carried out among samples of larvae from different localities for Mycobacterium, Sulfurospirillum, Desulfovibrio, Enhydrobacter, Aminomonas, which are associated with the eutrophication process and/or places rich in organic matter. There was a significant difference (p < 0.05) only to Desulfovibrio, Enhydrobacter, and Aminomonas. For these taxa, the Dunn test was performed, and the results are in S7 Table.” (Lines 223-227).

We added the following information in the discussion: “Like in Rojas et al. [58] variation in bacteria was expected among the larval samples from distinct regions of the same study area. The dissimilarity was demonstrated by the results of PERMANOVA analysis that showed the bacterial diversity in the larvae was considerably different among locations. Other studies exhibited the significant effect of breeding sites on the bacterial communities of mosquitoes during the larval stage [59]. The analysis of statistics revealed variations between larvae from different sites in terms of Desulfovibrio and Enhydrobacter taxa (S7 Table). Samples collected from the RO37 region differed from those of RO21, RO23 and RO7 regarding Desulfovibrio. Although other locations also have a strong odor because of the decomposition of organic matter, RO37 is close to an extensive field of soybean cultivation. This can lead to an increase in organic matter because of the release of products resulting from this activity or even the dragging of these products by rain, thus explaining the occurrence of this taxon and the difference found. Larvae samples that were collected in the nearest locality (RO7) to a settlement (Vila Teotônio) were the ones that showed a difference with those from the localities RO21, RO27, RO29, RO32 and RO37 in relation to Enhydrobacter. The presence of this taxon and the difference found can be explained by the proximity of the collection site to the settlement, which can have led to the increase of organic matter from anthropic activities. Although the larval samples showed bacterial genera associated with eutrophication with significant difference between locations, no bacterial genera that can be a biomarker to monitor organic matter in freshwater was stated in the LEfSe analysis.” (Lines 288-307).

In “Further studies are required to verify whether these genera can be used as biomarkers to monitor organic matter in freshwater” (lines 241-242) - perhaps the present study could apply analyzes (such as LDA and LEfSe) to detect differentially abundant bacteria (biomarkers) between areas with different classes of eutrophication.

Thank you for the comment. As mentioned in the previous answer, we performed the LEFSE analysis between samples of larvae from different locations and no taxon was indicated as a biomarker. We added the following information: “Although the larval samples showed bacterial genera associated with eutrophication with significant difference between locations, no bacterial genera that can be a biomarker to monitor organic matter in freshwater was stated in the LEfSe analysis” (Lines 305-307).

In the same way, the affirmation “However, the bacterial composition Mansonia larvae can serve as an alternative indicator of water quality when floating macrophytes are abundant” (lines 259-261) needs a better support from the study data - how did larvae microbiota vary according to water quality?

Thank you for this comment. We agree that better support is needed for the above information, as no association was made between water parameters and larval microbiota in this study. So we change the sentence to: “Therefore, studies need to be performed to verify if the bacterial composition of Mansonia larvae can serve as an alternative indicator of water quality when floating macrophytes are abundant.” (Lines 322-324).

In “Despite the lack of differences between the bacterial compositions of larvae from different breeding sites in the PCoA” (lines 262-263) - a better description of collection sites should be presented, describing similarities between breeding sites. This information should be considered in statistical analysis. It may help to understand if microbiota diversity and composition vary according to some environmental characteristic, such as degrees of human impact or presence of areas flooded by the hydroelectric plant, etc.

Thank you for this comment. Information about each larvae collection site has been added in S1 Table. Although it is not possible to visualize the differences in the PCoA, the PERMANOVA analysis showed that the bacterial diversity in the larvae was significantly different among many localities.

Collection in different sites could contribute to differences seen in the microbiota of different life stages and must be considered in the discussion.

Thank you for observation. We added the following information in the introduction: “Bascuñán et al. [22] confirmed that the water samples from breeding sites have more bacterial diversity than the microbiota of larvae, indicating that some bacteria in the water do not have contact with the larvae or cannot survive in their intestine. Much of the bacteria found in the intestine of adult mosquitoes are from the larval environment, and the variation in bacterial diversity between adult mosquitoes and larvae may be because of changes in diet or intestinal restructuring during metamorphosis [22]. The microbiota composition in mosquitoes is highly influenced by their geographic location. This is because each region adds different environmental conditions that can affect the variation of the microbiota, such as the amount of nutrients available in the larval habitat, microclimatic factors, and the availability of blood sources [23-25].” (Lines 57-66).

We also added the following information in the discussion: “Like in Rojas et al. [58] variation in bacteria was expected among the larval samples from distinct regions of the same study area. The dissimilarity was demonstrated by the results of PERMANOVA analysis that showed the bacterial diversity in the larvae was considerably different among locations. Other studies exhibited the significant effect of breeding sites on the bacterial communities of mosquitoes during the larval stage [59].” (Lines 288-292).

---

## [Decision Letter · Decision Letter 1]

25 Sep 2023

PONE-D-23-18212R1Bacterial diversity on larval and female Mansonia spp. from different localities of Porto Velho, Rondonia, Brazil.PLOS ONE

Dear Dr. Oliveira,

Thank you for submitting your manuscript to PLOS ONE. After careful consideration, we feel that it has merit but does not fully meet PLOS ONE’s publication criteria as it currently stands. Therefore, we invite you to submit a revised version of the manuscript that addresses the points raised during the review process.

We look forward to receiving your revised manuscript.

Kind regards,

Ahmed Ibrahim Hasaballah

Academic Editor

PLOS ONE

Journal Requirements:

Reviewers' comments:

Reviewer's Responses to Questions

**Comments to the Author**

1. If the authors have adequately addressed your comments raised in a previous round of review and you feel that this manuscript is now acceptable for publication, you may indicate that here to bypass the “Comments to the Author” section, enter your conflict of interest statement in the “Confidential to Editor” section, and submit your "Accept" recommendation.

Reviewer #1: All comments have been addressed

Reviewer #2: All comments have been addressed

2. Is the manuscript technically sound, and do the data support the conclusions?

Reviewer #1: Partly

Reviewer #2: Yes

3. Has the statistical analysis been performed appropriately and rigorously? 

Reviewer #1: Yes

Reviewer #2: Yes

4. Have the authors made all data underlying the findings in their manuscript fully available?

Reviewer #1: Yes

Reviewer #2: Yes

5. Is the manuscript presented in an intelligible fashion and written in standard English?

Reviewer #1: Yes

Reviewer #2: No

6. Review Comments to the Author

Reviewer #1: 6. The larvae were not identified beyond genus level. However, the adult females were. The authors need to use the extracted DNA to sequence and identify the larvae at species level. Mosquito larvae of different species can have significant differences in their microbiota even if they are reared in the same breeding site. Thus, treating all the larvae as Mansonia spp. and comparing them to adult females of different Mansonia species is not a good variable for the microbiota analyses.

Thank you for this comment. We agreed on the importance of identifying the larvae at the species level, however, unfortunately we were unable to carry out this step, as there is no more genetic material from the larval specimens. Analyzes of the bacterial diversity between larvae and adults were carried out considering only the genus, without considering the species of the adults.

I strongly encourage the authors to add this point to the manuscript as one of the limitations of the study.

Reviewer #2: I am happy with the author's corrections of the manuscript edits based on my recommendations. It is clearer to follow, and data is now deeper explored. I have some additional comments which can be addressed easily. Overall, I believe that it still needs some English revision as some phrases are hard to comprehend (indicated below).

Introduction:

Lines 49-51: This text is repetitive - “Mosquitoes contain a range of symbiotic microorganisms that are crucial to their development [10]. The interactions between mosquitoes and bacteria are crucial for larval development [11].”

Lines 55-57: Please rephrase to “These microorganisms can be obtained by the larvae from the environment, primarily from water in aquatic habitats, and vertically [19-21].

Lines 67-68: Please rephrase to “Symbiotic bacteria can be used as a potential tool for both mosquito and pathogen control.”

Lines 69-70: The phrase “As the composition of microbiota in adult mosquitoes is derived from their larval stages and water of the breeding [21], changes in the larval microbiota impact not only larval but also adult development” is confusing - what do the authors mean by “adult development”?

Lines 79-82: I suggest “verify differences in bacterial composition associated with the larvae and females between areas in the vicinities of Santo Antonio Energia hydroelectric power plant (SAE) reservoir in the municipality of Porto Velho, Rondonia state, Brazil”.

Methods:

Line 89: Please clarify if larvae and adult mosquitoes were collected in the same period.

Results:

Please indicate female species in Table 1 (could be inside parentheses as it is shown for plant species).

Discussion:

The collection of larvae and adults in different places might have contributed to differences seen in the microbiota between different life stages. This must be considered in the discussion.

Lines 253-255: Please rephrase to “Although Mansonia mosquitoes are not considered the primary vectors of arboviruses, some were found naturally infected with the Western Equine Encephalitis virus in Argentina”.

Line 262-265: Please rephrase to “bacterial communities observed during the larvae stage of Mansonia species may vary from those of culicids with distinct ecology”.

Line 266-267: What do the authors mean by “a connected bacterial grouping”?

Lines 268-269: Please rephrase to “The frequency of Desulfovibrio in larvae (3.59%) was higher than in adults (0.005%).”

Line 269: The 16S sequencing allow the comparison of bacteria relative abundance between groups of samples, so it is not possible to say that “Mansonia spp. the larvae hosted a larger quantity of the sulfur-reducing bacteria of the genus Sulfurospirillum” (please modify to “larger relative amount” or similar). The same is true for “… were found in large numbers” (line 271).

Line 276: Please rephrase to “The vegetation root microhabitat fosters anaerobic bacteria, such as Aminomonas”.

Line 333: What do the authors mean by “insect populations”? Maybe you mean “insect species”.

7. PLOS authors have the option to publish the peer review history of their article (what does this mean?). If published, this will include your full peer review and any attached files.

Reviewer #1: No

Reviewer #2: No

---

## [Author Response · Author response to Decision Letter 1]

19 Oct 2023

Reviewer #1: 

6. The larvae were not identified beyond genus level. However, the adult females were. The authors need to use the extracted DNA to sequence and identify the larvae at species level. Mosquito larvae of different species can have significant differences in their microbiota even if they are reared in the same breeding site. Thus, treating all the larvae as Mansonia spp. and comparing them to adult females of different Mansonia species is not a good variable for the microbiota analyses.

Thank you for this comment. We agreed on the importance of identifying the larvae at the species level, however, unfortunately we were unable to carry out this step, as there is no more genetic material from the larval specimens. Analyzes of the bacterial diversity between larvae and adults were carried out considering only the genus, without considering the species of the adults.

I strongly encourage the authors to add this point to the manuscript as one of the limitations of the study.

Thank you for this observation. We added the following text: "It is largely known that geographic location, aquatic ecosystem biotic and abiotic factors, and mosquito species can influence on the composition of the microbiota [23,25,59]. Thus, this study faces a limitation in understanding the bacterial difference found between the adults and larvae because the later life stages were identified to the genus level. Likely, geographic location and the mosquito species may have contributed to the differences observed in the bacterial composition of the larvae and females." (Lines 327-332).

Reviewer #2: 

Introduction:

Lines 49-51: This text is repetitive - "Mosquitoes contain a range of symbiotic microorganisms that are crucial to their development [10]. The interactions between mosquitoes and bacteria are crucial for larval development [11]."

Thank you for this comment. We changed the text to "Mosquitoes contain a range of symbiotic microorganisms that are crucial to their development [10]. The interactions between mosquitoes and bacteria during larval development can stimulate metabolic changes with functional consequences on adult fitness [11]." (Lines 50-53).

Lines 55-57: Please rephrase to "These microorganisms can be obtained by the larvae from the environment, primarily from water in aquatic habitats, and vertically [19-21].

Thank you for this observation. The sentence was reformulated according to your suggestion: "These microorganisms can be obtained by the larvae from the environment, primarily from water in aquatic habitats, and vertically [19-21]." (Lines 57-59).

Lines 67-68: Please rephrase to "Symbiotic bacteria can be used as a potential tool for both mosquito and pathogen control."

Thank you for your kind attention. The text was rewritten as suggested: "Symbiotic bacteria can be used as a potential tool for both mosquito and pathogen control [26,27]." (Lines 69-70).

Lines 69-70: The phrase "As the composition of microbiota in adult mosquitoes is derived from their larval stages and water of the breeding [21], changes in the larval microbiota impact not only larval but also adult development" is confusing - what do the authors mean by "adult development"?

Thank you for your careful attention. We wanted to refer to adult fitness. We changed the text to "As the composition of microbiota in adult mosquitoes is derived from their larval stages and water of the breeding [21], changes in the larval microbiota can impact not only larval development but also adult fitness [10,11]." (Lines 70-72).

Lines 79-82: I suggest "verify differences in bacterial composition associated with the larvae and females between areas in the vicinities of Santo Antonio Energia hydroelectric power plant (SAE) reservoir in the municipality of Porto Velho, Rondonia state, Brazil".

Thank you for this suggestion. The sentence was reformulated according to your suggestion: "verify differences in bacterial composition associated with the larvae and females between areas in the vicinities of Santo Antonio Energia hydroelectric power plant (SAE) reservoir in the municipality of Porto Velho, Rondonia state, Brazil." (Lines 81-83).

Methods:

Line 89: Please clarify if larvae and adult mosquitoes were collected in the same period.

Thank you for this observation. We added the following sentence: "Collections were conducted approximately every four months from October 2018 to March 2020 (S2 Table)." (Lines 89-91).

Results:

Please indicate female species in Table 1 (could be inside parentheses as it is shown for plant species).

Thank you for this suggestion. We added the names of the analyzed species at each ID locality in Table 1 and changed the footnote to: "Value inside parentheses in “Host plant species" column shows the number of larvae analyzed from each host plant species in each locality." (Lines 188-189).

Discussion:

The collection of larvae and adults in different places might have contributed to differences seen in the microbiota between different life stages. This must be considered in the discussion.

Thank you for this observation. We added the following text: "It is largely known that geographic location, aquatic ecosystem biotic and abiotic factors, and mosquito species can influence on the composition of the microbiota [23,25,59]. Thus, this study faces a limitation in understanding the bacterial difference found between the adults and larvae because the later life stages were identified to the genus level. Likely, geographic location and the mosquito species may have contributed to the differences observed in the bacterial composition of the larvae and females." (Lines 327-332).

Lines 253-255: Please rephrase to "Although Mansonia mosquitoes are not considered the primary vectors of arboviruses, some were found naturally infected with the Western Equine Encephalitis virus in Argentina".

Thank you for this comment. We followed your suggestion and now it reads: "Although Mansonia mosquitoes are not considered the primary vectors of arboviruses, some were found naturally infected with the Western Equine Encephalitis virus in Argentina [39]." (Lines 255-257).

Line 262-265: Please rephrase to "bacterial communities observed during the larvae stage of Mansonia species may vary from those of culicids with distinct ecology".

Thank you for this observation. The text was changed as suggested: "the bacterial communities observed during the larvae stage of Mansonia species may vary from those of culicids with distinct ecology." (Lines 264-266).

Line 266-267: What do the authors mean by "a connected bacterial grouping"?

Thank you for this observation. This sentence was not clear and we changed the text to: "The bacterial genus Desulfovibrio contains gram-negative sulfate-reducing bacteria, which are present in organic-rich environments and are linked to water bio-corrosion [47]. This bacterial genus has been identified in the larvae of Mansonia spp. studied and other insects [48,49]." (Lines 266-269).

Lines 268-269: Please rephrase to "The frequency of Desulfovibrio in larvae (3.59%) was higher than in adults (0.005%)."

Thank you for this comment. The sentence was rewritten and now reads is: "The frequency of Desulfovibrio in larvae (3.59%) was higher than in adults (0.005%)." (Lines 269-270).

Line 269: The 16S sequencing allow the comparison of bacteria relative abundance between groups of samples, so it is not possible to say that "Mansonia spp. the larvae hosted a larger quantity of the sulfur-reducing bacteria of the genus Sulfurospirillum" (please modify to "larger relative amount" or similar). The same is true for "… were found in large numbers" (line 271).

Thank you for this observation. We changed the text to: "Besides Desulfovibrio, Mansonia spp. larvae hosted a larger relative amount of the sulfur-reducing bacteria of the genus Sulfurospirillum [50]. The Aminomonas bacteria from the Synergistaceae family were found in large relative amount in the larvae, however, no contigs were recognized in Mansonia females." (Lines 270-274).

Line 276: Please rephrase to "The vegetation root microhabitat fosters anaerobic bacteria, such as Aminomonas".

Thank you for this observation. We changed the sentence as suggested: "The vegetation root microhabitat fosters anaerobic bacteria, such as Aminomonas." (Lines 277-278).

Line 333: What do the authors mean by "insect populations"? Maybe you mean "insect species".

Thank you for this comment. Exact, like "insect species". We changed to: "Estimated that approximately 65% of the insect species are infected with Wolbachia [65]." (Line 342).

---

## [Editor Report · Decision Letter 2]

23 Oct 2023

Bacterial diversity on larval and female Mansonia spp. from different localities of Porto Velho, Rondonia, Brazil.

PONE-D-23-18212R2

Dear Dr. Oliveira,

We’re pleased to inform you that your manuscript has been judged scientifically suitable for publication and will be formally accepted for publication once it meets all outstanding technical requirements.

Kind regards,

Ahmed Ibrahim Hasaballah

Academic Editor

PLOS ONE
---

## [Editor Report · Acceptance letter]

15 Nov 2023

PONE-D-23-18212R2 

Bacterial diversity on larval and female *Mansonia* spp. from different localities of Porto Velho, Rondonia, Brazil. 

Dear Dr. Oliveira:

I'm pleased to inform you that your manuscript has been deemed suitable for publication in PLOS ONE. Congratulations! Your manuscript is now with our production department. 

Kind regards, 

on behalf of

Dr. Ahmed Ibrahim Hasaballah 

Academic Editor

PLOS ONE